# Effects of Processing on Starch Structure, Textural, and Digestive Property of “Horisenbada”, a Traditional Mongolian Food

**DOI:** 10.3390/foods11020212

**Published:** 2022-01-13

**Authors:** Hongyan Li, Zhijun Chen, Yifan Mu, Ruolan Ma, Laxi Namujila, Minghai Fu

**Affiliations:** 1China-Canada Joint Lab of Food Nutrition and Health (Beijing), School of Food and Health, Beijing Technology and Business University (BTBU), 11 Fucheng Road, Beijing 100048, China; hongyan.li@btbu.edu.cn (H.L.); chenzhijun0620@163.com (Z.C.); muyifan1998@163.com (Y.M.); maruolanya@163.com (R.M.); 2NMPA Key Laboratory of Quality Control of Traditional Chinese Medicine (Mongolian Medicine), Inner Mongolia University for Nationalities, Tongliao 028000, China; namujila@126.com; 3School of Mongolian Medicine, Inner Mongolia University of Nationalities, Tongliao 028000, China

**Keywords:** millet, whole grain, molecular structure, digestibility, texture

## Abstract

Horisenbada, prepared by the soaking, steaming, and baking of millets, is a traditional Mongolian food and is characterized by its long shelf life, convenience, and nutrition. In this study, the effect of processing on the starch structure, textural, and digestive property of millets was investigated. Compared to the soaking treatment, steaming and baking significantly reduced the molecular size and crystallinity of the millet starch, while baking increased the proportion of long amylose chains, partially destroyed starch granules, and formed a closely packed granular structure. Soaking and steaming significantly reduced the hardness of the millets, while the hardness of baked millets is comparable to that of raw millet grains. By fitting digestive curves with a first-order model and logarithm of the slope (LOS) plot, it showed that the baking treatment significantly reduced the digestibility of millets, the steaming treatment increased the digestibility of millets, while the soaked millets displayed a similar digestive property with raw millets, in terms of both digestion rate and digestion degree. This study could improve the understanding of the effects of processing on the palatability and health benefits of Horisenbada.

## 1. Introduction

Horisenbada, prepared by the soaking, short-time steaming, and high temperature baking of millets, is a traditional Mongolian food. Horisenbada is usually consumed as porridge with tea or yogurt, having good palatability, strong satiety, and a unique flavor. It is an important daily staple-food for Mongolian steppe herdsmen due to its long shelf life, convenience, and nutrition. Millets are one of the major cereal grains, mainly distributed in arid and semi-arid areas of Africa and Asia (China and India) [1]. In general, millets are enriched in essential nutrients, such as protein, fat, carbohydrates, minerals, vitamins, and bioactive compounds [2]. Phenolic acid, flavonoids, and other bioactive compounds in millets exhibit multiple health benefits, including antioxidant and anti-microbial activities [3]. Besides, millets can be processed in various ways including cooking, fermentation, toasting, puffing, etc. [4], which could contribute to the alteration of millet structure, texture, and digestion properties [5,6].

During the manufacture of Horisenbada, soaking, steaming, and baking processes are of prime importance. Soaking is a basic pretreatment widely used in cereal processing. For instance, the presoaking before rice cooking allowed water molecule to diffuse into the inside of the rice kernel [7]. Presoaking made grains easier to be gelatinized in the later processing [8]. In addition, alkali soaking significantly increased the amylopectin amount in the leachate, which increased the stickiness of cooked rice [9], whereas soaking in diverse solutions, such as water, sodium chloride (1%, *w*/*v*), or sodium bicarbonate (0.75%, *w*/*v*), did not affect the in vitro starch digestion [10].

Steaming is one of the most common grain processing methods. A series of previous research proved that cooking method [11], steaming time [12], and steaming temperature [13] can influence the texture of cooked grains, which was essentially related to the gelatinization and molecular structure of grains [14]. Our previous study found that rice grains during parboiling might lead to a less sticky texture due to starch gelatinization in the surface layer of cooked rice via blocking starch leaching [15]. Furthermore, the stickiness between cooked rice grains was strongly affected by the molecular structure of the leached starch [16]. One study also reported that cooked rice’s morphological structure and in vitro digestion rate was varied by cooking method [17].

Baking is a dry heating process widely used in the manufacture of grain foods. It has been proven that baking enhanced the flavor by browning the surface of millet grains and increasing the variety of aroma compounds to give millets a specific odor [18]. In addition to changing the flavor properties, dry heating also changed the molecular size of the grain starch and reduced the long-amylose chains with the degree of polymerization (DP) ~5000–20,000 [19]. One also found that the baking process obviously decreased the digestibility of grain starch, which was associated with the aggregation of compact starch granules [20]. Considering that the production of Horisenbada mainly involves soaking, steaming, and baking processes, it is reasonable to propose that the positive effects of processing on the textural and digestible properties of Horisenbada might be strongly associated with the alteration of the structural changes of millet starch. So far, the effects of soaking, steaming, and baking on the starch structure, texture, and digestion of millet grains are not known. Therefore, the association between millet starch structure (molecular structure, crystal structure, and granule structure) and functional properties (hardness, digestion) were explored in this study.

## 2. Materials and Methods

### 2.1. Materials

Three millet cultivars harvested from different areas in inner Mongolia Chifeng, Ordos, and Tongliao, respectivey, with a known starch content of 70.7%, 69.0%, and 79.1%, respectively, were used and denoted as M1, M2, and M3, respectively. Dimethyl sulfoxide (DMSO, HPLC grade for analysis) was obtained from Merck Co. Inc. (Darmstad, Germany). Protease (type XIV, from Streptomyces griseus), Pepsin (672 U/mg, from porcine gastric mucosa), α-amylase (≥5 U/mg, from porcine pancreas), and amyloglucosidase (≥260 U/mL, from Aspergillus niger) were purchased from Sigma-Aldrich Pty (St. Louis, MO, USA). Isoamylase (200 U/mL) and GOPOD assay kit were purchased from Megazyme Ltd. (Wicklow, Ireland). All other reagents were of analytical grade.

### 2.2. Preparation of Horisenbada

Ten grams of dehulled millets were soaked in 50 mL of distilled water at 35 °C for 15 min, and then the water was removed. Further, millets were steamed-cooked for 4 min without extra water, then laid on a baking tray and rapidly baked in an oven at 180 °C for 13 min. For comparison, the millets sample, soaked in 50 mL of distilled water at 35 °C for 15 min and then dried at 60 °C for 12 h, was nominated as the “Soaked” sample; the one soaked in 50 mL of distilled water at 35 °C for 15 min, steamed-cooked for 4 min, and then dried at 60 °C for 12 h, was nominated as the “Steamed” sample; the counterpart, processed with the same procedures of *Horisenbada*, was nominated as the “Baked” sample.

### 2.3. Molecular Size Distribution

The determination of molecular size distribution was conducted following the method described elsewhere [21]. Eight milligrams of millet powder were treated by protease and sodium bisulfite, and then precipitated by adding 10 mL of ethanol. The sample was dissolved in a DMSO solution with 0.5% (*w*/*w*) LiBr (DMSO/LiBr). A size exclusion chromatography system consisted of GRAM 30, GRAM 3000 columns (PSS, Mainz, Germany) and an RID-10A refractive index detector.

### 2.4. Chain Length Distribution

The deproteinization treatment was consistent with Section 2.3. Further, the sample was dissolved in 0.9 mL of hot deionized water. After cooling to room temperature, 0.1 mL of acetate buffer at pH 3.7 and 6.25 μL isoamylase were added for starch debranching. The mixed solution was incubated at 37 °C for 3 h and then heated at 80 °C for 2 h. The debranched starch sample was freeze-dried and ultimately dissolved in the DMSO solution with 0.5% (*w*/*w*) LiBr (DMSO/LiBr) for Size-Exclusion Chromatography (SEC) separation with GRAM100 and GRAM1000 columns [22].

### 2.5. X-ray Diffraction (XRD)

The X-ray diffractometer (D2 PHASER, Bruker AXS GMBH, Karlsruhe, Germany) with Cu Kα radiation (λ = 0.154) was used to record XRD patterns. The scanning range was 5°–40° at a rate of 2°/min with a scanning step of 0.02. Before the operation, the moisture content of the samples was equilibrated to about 10% by standing at room temperature overnight. The crystallinities of the samples were measured with HighScore Plus 5.1 software (Malvern Panalytical Ltd., Malvern, UK).

### 2.6. Scanning Electron Microscopy (SEM)

The morphology of the millet particles was obtained using a JSM-7610FPlus SEM (JEOL Ltd., Tokyo, Japan). Samples were first fixed on an aluminum stub before gold sputtering. Then, the millet was imaged by the scanning electron microscope at an accelerating voltage of 15 kV.

### 2.7. Textural Profile Analysis (TPA)

The determination was carried out according to the method by Li et al. [15]. After cooling the millets to room temperature, single whole millet grain was placed on the base plate. The target distance test was used for measurements with a Texture analyzer (Brookfield Engineering Laboratories, Middleboro, MA, USA) with a TA5 probe attachment. The compression settings were as follows: target value, 0.9 mm; trigger load, 1 g; test speed, 0.70 mm/s. After the compression test, the value of hardness was recorded by TexturePro CT software. Each sample was performed 20 times.

### 2.8. Millet Digestion

Millet grains were ground and sifted through 80 mesh. Millet powder samples containing 90 mg of starch were firstly cooked in a 50 mL centrifuge tube with 6.0 mL of deionized water at 100 °C for 30 min and then cooled to 37 °C in a water bath. A 5.0 mL pepsin solution (1 mg/1 mL 0.02 mol HCI) was added to the samples. Meanwhile, 5.0 mL of 0.02 mol HCI were added to the controls. After incubation at 37 °C for 30 min, all sample solutions were neutralized by 5 mL of 0.02 mol NaOH. In quick succession, 5.0 mL of porcine α-amylase/amyloglucosidase mixture enzyme (135 U porcine α-amylase and 1 U amyloglucosidase in 5 mL 0.2 mol sodium acetate buffer at pH 6) were added to samples incubated in a water bath at 37 °C and stirred with a magnetic stirrer bar at 300 rpm. Afterwards, 100 μL aliquots were transferred and dispersed into 900 μL of absolute ethanol to terminate the reaction at a series of time points. Then, 100 μL of digestion solution were added to 3.0 mL of GOPOD reagent (glucose oxidase/peroxidase determination reagent). All samples were incubated at 50 °C for 20 min. Then, a 200 μL solution was added to 96-pour plates and measured at the absorbance of 510 nm by a Synergy H1 microplate reader (Biotek lnc, Winooski, VT, USA). The digestibility was calculated according to reference [23] using the following Equation (1):(1)$$\%\mathrm{Digested}=\Delta A\left(\mathrm{Sample}\right)\times \frac{100\;\mu L\times 1.0\;\mathrm{mg}/\mathrm{mL}}{\Delta A\left(D-\mathrm{Glucose}\;\mathrm{Standard}\right)}\times 10\times 210\times \frac{100\;\%}{90\;\mathrm{mg}}\times \frac{162}{180}$$
where ∆A (Sample) is the absorbance at each time point and ∆A (D-Glucose Standard) is the absorbance from the standard D-glucose solution. The value 10 × 210 and 162/180 is the computational multiple from 100 μL aliquots to 21.0 mL reaction solution and the transformation coefficient from starch to glucose in weight, respectively.

### 2.9. Fitting to First-Order Kinetics

Starch digestion data can be fitted to a first-order Equation (2):(2)$$C_{t}=C_{\infty }\left(1-e^{-kt}\right)$$

Then the Equation (2) can be transformed into LOS plot where there is a linear relationship between ln(dC_t_/dt) and *k*, using the following Equation (3)
(3)$$\mathrm{In}\left({\mathrm{dC}}_{t}/\mathrm{dt}\right)=-kt+\mathrm{In}\left(C_{\infty }k\right)$$

The value of *k* and *C*_∞_ are calculated from the slope and intercept, which represent −*k* and ln(*C_∞_k*), respectively. In this study, the slope was estimated from the second-order finite-difference formula ln[(Ci+1-Ci-1)/(ti+1-ti-1)] as functions of (ti+1-ti-1)/2 for all points except the first and last point [24].

### 2.10. Statistical Analysis

All the data was analyzed with analysis of variance (ANOVA) with Tukey’s pairwise comparisons for a statistical significance and all values were expressed as mean ± standard deviation. SPSS 22.0 software (SPSS Inc, Chicago, IL, USA) was used for the above data analysis.

## 3. Results

### 3.1. Molecular Size of Millet Starch

#### 3.1.1. Molecular Size Distribution of Branched Starch

Typical SEC weight molecular distributions of wholly branched millet starch are presented in Figure 1. The average R_h_, denoted by $\bar{R_{h}}$, was calculated and shown in Table 1. The molecular size of branched starch was mainly distributed at Rh~1–1000 nm [25]. Two populations of branched starch with different molecular size could be found: amylose (AM, 10 < R_h_ ≤ 100 nm) and amylopectin (AP, R_h_ > 100 nm) [26]. As shown in Figure 1, the proportion of starch molecules with R_h_ > 100 nm for steamed and baked millet starch was decreased. In Table 1, the $R_{h}$ of steamed and baked millet starch was significantly reduced, especially $R_{h}$ of baked millet starch decreasing to less than 10 nm. For the three millet cultivars, there was the same trend between samples as affected by different processes, showing that steaming and baking treatments significantly reduced the molecular size of the millet starch, especially for the baking process. This also indicates that the steaming and baking process might cause the degradation of starch molecules, especially amylopectin. One study on maize starch also supported that a high temperature could cause the degradation of long starch branches [19].

#### 3.1.2. CLDs of Debranched Starch

Figure 2 presents typical CLDs of debranched starch. Starch CLDs, *w*_(logX)_, obtained from the DRI signal were plotted against DP *X*. As described elsewhere [15], the population of chains with *X* ≤ 100 and 100 < *X* ≤ 10,000 were defined as AP chains and AM chains, respectively. For the amylopectin CLDs, it showed the usual features of two large amylopectin peaks, while for the amylose CLDs, the millet starch also displayed two peaks with one at *X*~200–300 and the other at *X*~700 [27]. Clearly, raw and soaked starch had similar CLD profiles, while steamed and baked starch had a higher peak in the amylose region, particularly for baked starch, displaying an extremely high peak. As shown in Table 1, the amylose region could be further divided into three groups: 100 < *X* ≤ 1000, 1000 < *X* ≤ 5000, and 5000 < *X* ≤ 20,000, which were defined as short, medium, and long amylose chains, respectively [28]. Interestingly, compared to raw and soaked starch, the proportion of short-amylose chains with *X*~100–1000 of baked and steamed starch was significantly increased, especially for that of baked starch, which almost doubled.

Compared with raw and other processed millet, the chain length distribution of baked millet starch had a higher proportion of chains with *X*~100–1000 and *X*~1000–5000. This might be caused by the degradation of long amylose chains and/or the repolymerization of short starch chains [19]. It was reported that a part of the medium- and long-amylose chains were depolymerized under a high temperature, thereby generating short chains with *X*~100–1000 and *X* ≤ 100. On the other hand, with the heating temperature continuously increasing, the short amylopectin chains were also repolymerized by forming new glycosidic bonds and presented a nonlinear structure with a higher hydrodynamic radius [29], contributing to the apparent increment of starch chains with *X*~100–1000.

### 3.2. The Crystalline Structure of Different Processed Millets

The XRD patterns and relative crystallinity of starches from raw and different processed millets are illustrated in Figure 3. Compared to these three millet cultivars, the crystallinity of raw millet starch was different, which might be related to the different amylose content. All raw and processed millet starch displayed the typical A-type diffraction pattern, which includes a doublet peak around 2θ~17°and 18°, and two single peaks 2θ~15°and 30° [30]. Obviously, for these three millet cultivars, the soaking treatment did not change the relative crystallinity of the millet starch while the steaming and baking processes significantly reduced the peak intensities of the millet starch, especially for the baked millet starch displaying the lowest relative crystallinities. This indicated that the steaming and baking processes made the millet starch gelatinized to some extent.

### 3.3. The Granule Structure of Different Processed Millets

The morphologies of raw and different processed millets were analyzed using SEM. As shown in Figure 4, the raw, soaked, and steamed millet starch of M1 displayed a spherical shape, whereas the granular structure of raw, soaked, and steamed millet starch of M2 and M3 was polygonal shaped. Although steaming made the millet starch gelatinized, the granular morphology was still observed by SEM, which may be associated with short-time steaming. For the three millet cultivars, the baked millet starch granules were partially destroyed and aggregated into irregular lumps with a closely packed granule structure. This might be attributed to starch gelatinization, moisture loss, and inter-granules connections under the high temperature [31].

### 3.4. Textural Property of Different Processed Millets

As shown in Figure 5, all three millet cultivars showed significantly different hardness. This might be due to the variations in terms of amylose content and the proportion of amylose branches [32]. For the effect of processing on millet texture, all cultivars presented the similar trend that raw millet was generally harder than soaked and steamed millet, while the hardness of baked millet was close to that of raw millet. It was readily understood that the soaking process could soften the millet grains by increasing the moisture content, and steaming could decrease the hardness of the millet grains by gelatinizing the millet starch, whereas the baking treatment further dried the millet grains and increased the firmness of the millet accordingly. The result was consistent with the SEM results, indicating that millet starch aggregated into lumps under high temperature can increase the hardness of the millet particle [20,31]. Compared with millet sample M1 and M2, M3 had a small difference between different processes, which might be caused by the lower amylose content of M3 [21].

### 3.5. Millet Starch of Digestion

In vitro digestion curves of raw and different processed millets are shown in Figure 6. All curves present the similar trend that the initial digestion rate increased rapidly, and then slowed down until reaching equilibrium. Generally, raw and soaked millets displayed the overlapped digestion curve. Compared to other processed millet, steamed millet presented the highest digestibility due to the gelatinization of the starch granules [33]. Interestingly, though the baking process can make millet starch gelatinized as well, the digestion curve of baked millet exhibited the lowest digestibility. Based on the result of SEM, the baking process made starch granules packed tightly, plus, TPA also proved the baked millet possessing a higher mechanic property in terms of hardness, which could block the enzyme approaching and hydrolysis pathways. Further, the generation of nonlinear structures, indicated from starch CLD, could also reduce starch digestibility. In addition, the effect of processing on the digestion remained consistent between the three cultivars (Appendix A).

Further parameterizing the digestion between millet samples, LOS plots and model fitting curves are shown in Figure 7 and Table 1 (those for M2 and M3 were displayed in Appendix A). LOS plots proved that there was a linear correlation between digestion time and logarithmic form of digestion data. Meanwhile, first-order kinetics were used to fit the digestion curve and predict the reaction end-point of starch digestion. Two relevant parameters, *k* and *C*_∞_, were calculated from LOS plot. *k* is the starch digestion rate coefficient, and *C*_∞_ is the estimated percentage of starch digested at the end-point of the reaction. As shown in Table 1, steamed millet and baked millet presented the highest and lowest starch digestion rate, respectively. Meanwhile, the amount of digestible starch after the steaming process was the highest, while the quantity of digestible starch after the baking treatment was reduced to the minimum in terms of *C*_∞_ values. In addition, the *k* and *C*_∞_ values of raw and soaked millet had no significant differences, keeping consistent with the observation in Figure 6.

The soaking, steaming, and baking processes significantly changed the digestive property of the millet grains, while the variation trends remained consistent between all three cultivars. For details, the steamed millet showed the rapid digestive property, which was consistent with the highest *k* and *C*_∞_ values. The increased millet digestion may be caused by the gelatinization of millet starch by the steaming process. The soaking process did not significantly change the molecular, crystalline, and granular structure of the millets, which explains the similar digestion property between the raw and soaked millet. Interestingly, the baked millet displayed the most desirable digestive property in terms of the lowest *k* and *C*_∞_ values, whereas the molecular size and the degree of gelatinization of the baked millet starch were the lowest, indicating that the degree of gelatinization is not the only factor affecting cereal digestion. On one hand, it is reported that the baking process made starch granules packed tightly and blocked the enzyme approaching and hydrolyzing pathways [20], which was also supported by the results of the granular structure and hardness in this study. On the other hand, the newly formed glycosidic bonds inside the starch branches during the baking process might also contribute to the slow digestion of baked millets [19].

## 4. Conclusions

This study investigated the effect of various processes on starch structure, texture, and digestibility of millet during the preparation of “Horisenbada”. Between these different processing methods, the baking treatment affected the molecular structure, crystalline structure, and granular structure most significantly, which explained the obviously increased hardness and retarded digestibility. The possible links between starch structure, textural, and digestive properties of millets as affected by different treatments were also proposed, which provided fundamental evidence for Horisenbada as a nutritional food with low digestibility. The study could be beneficial to understanding the structural reasons for the palatability and heathy benefits of Horisenbada, the traditional Mongolian food.

## Figures and Tables

**Figure 1 foods-11-00212-f001:**
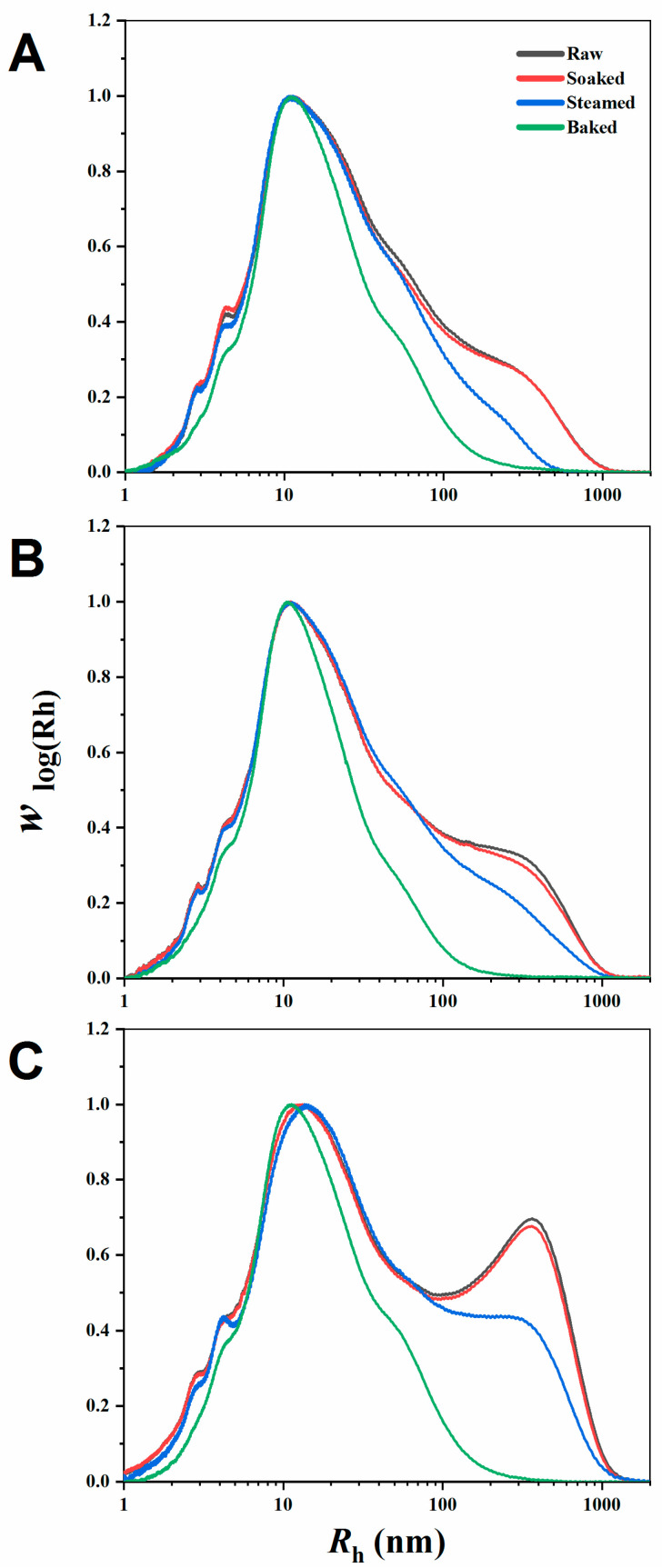
Molecular weight distribution of raw and different processed millet starch (soaked, steamed, and baked, respectively). (**A**–**C**) represents M1, M2, M3, respectively.

**Figure 2 foods-11-00212-f002:**
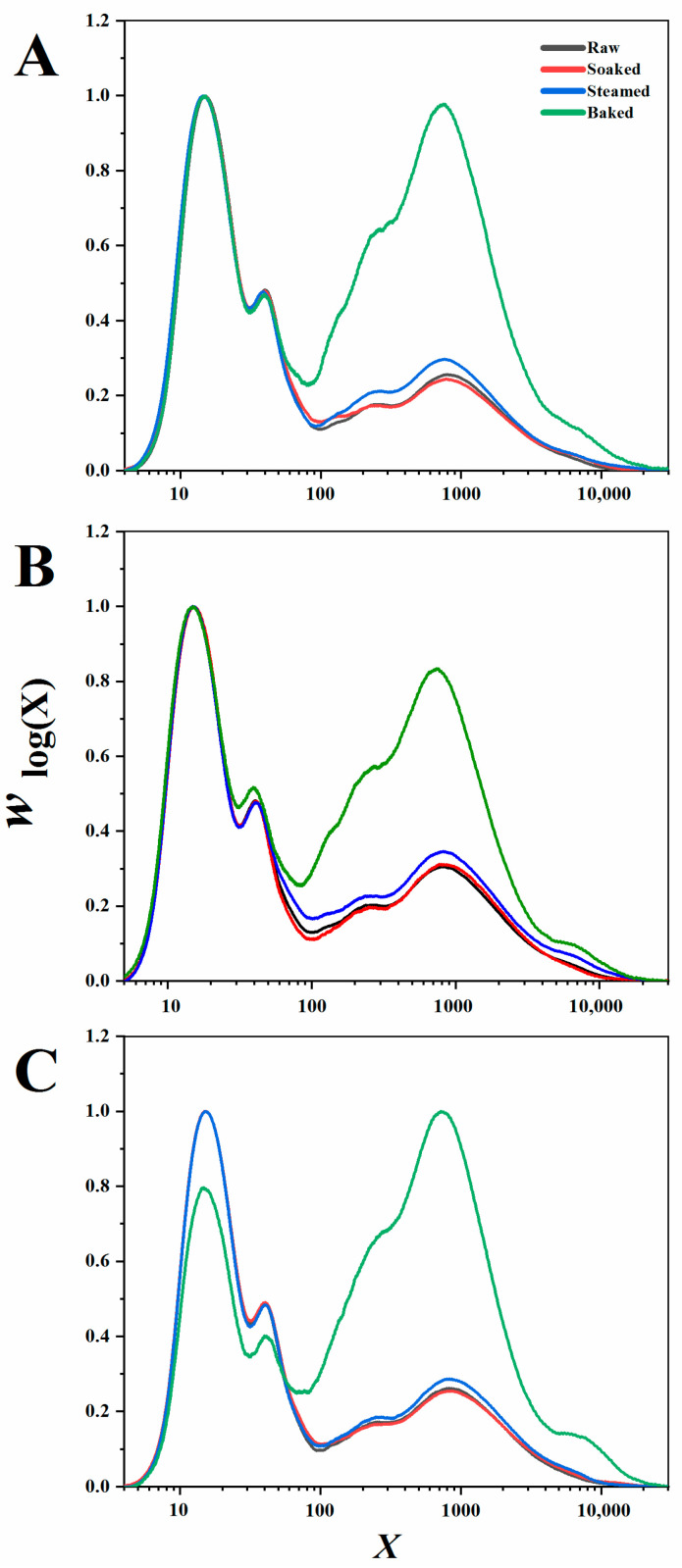
Chain length distributions of raw and different processed millet starch (soaked, steamed, and baked, respectively). (**A**–**C**) represents M1, M2, M3, respectively. X represented the DP value of an individual chain.

**Figure 3 foods-11-00212-f003:**
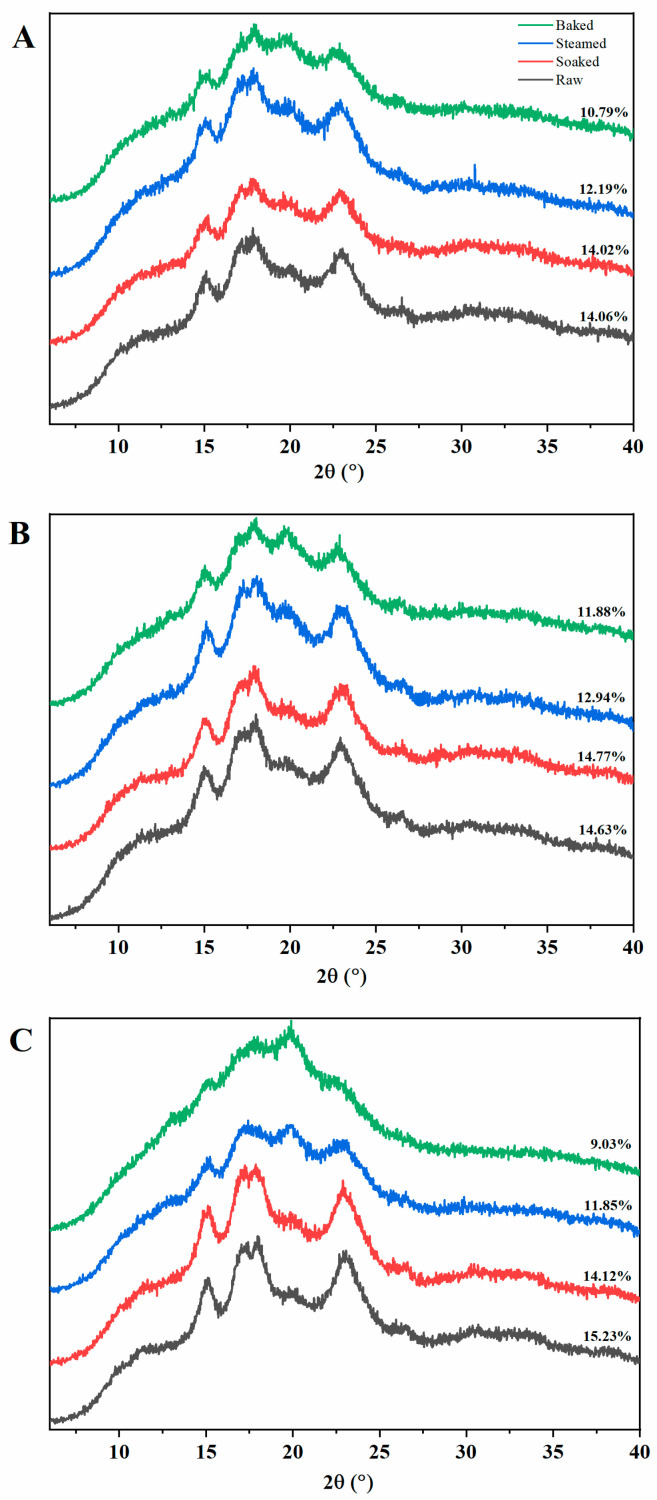
XRD patterns of raw and different processed starch from three cultivars of millet (**A**–**C**) represents M1, M2, M3, respectively.

**Figure 4 foods-11-00212-f004:**
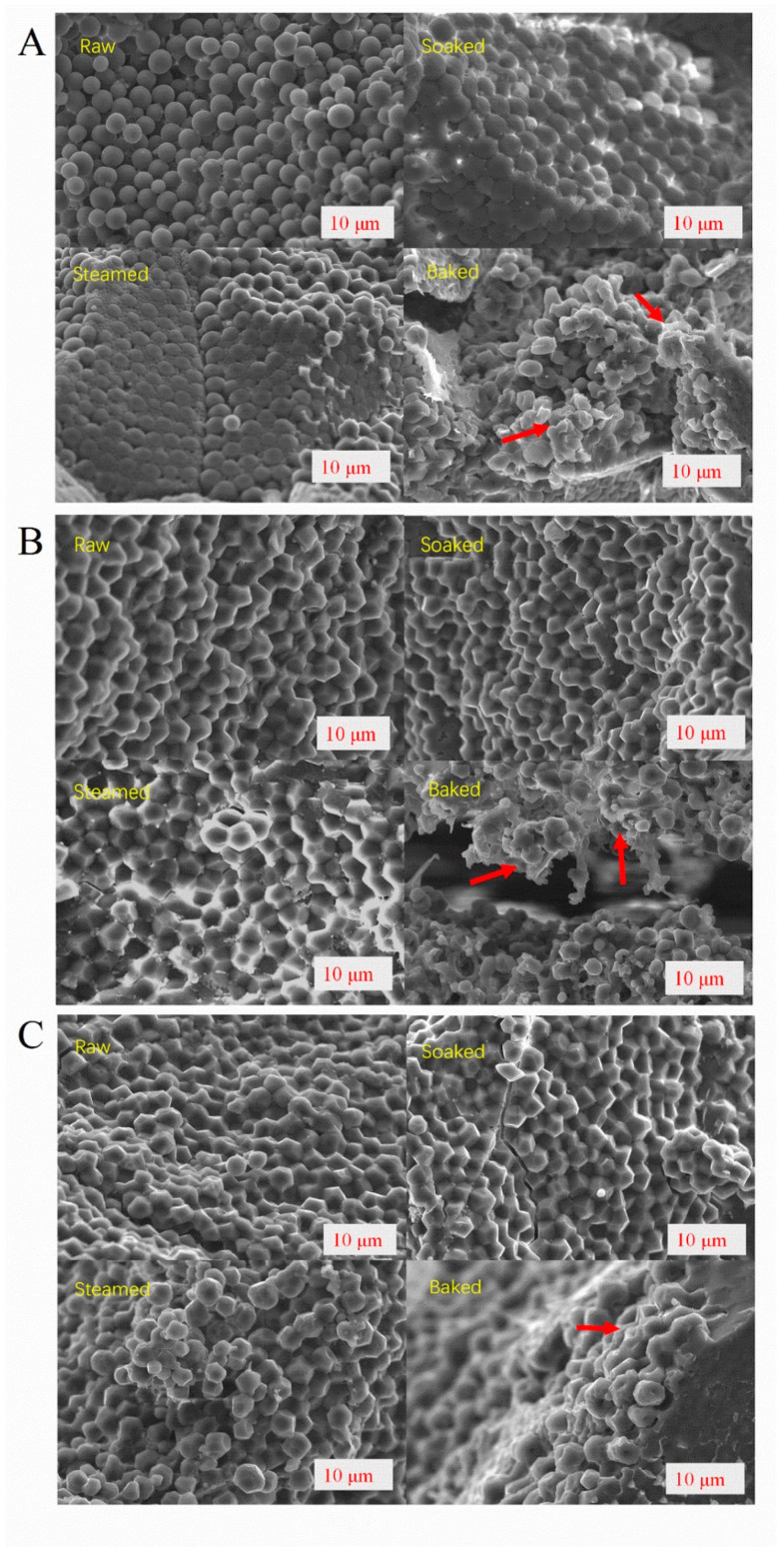
Scanning electron micrographs of raw and different processed millets at 1000× magnification with 10 μm scale bars. (**A**–**C**) represents M1, M2, M3, respectively.

**Figure 5 foods-11-00212-f005:**
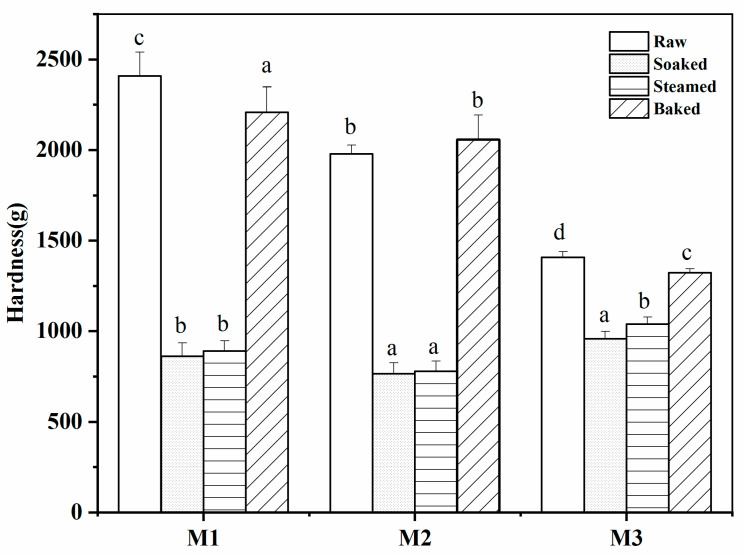
The hardness of raw and different processed millet grains. Columns with different letters for hardness for each millet are significantly different with *p* < 0.05.

**Figure 6 foods-11-00212-f006:**
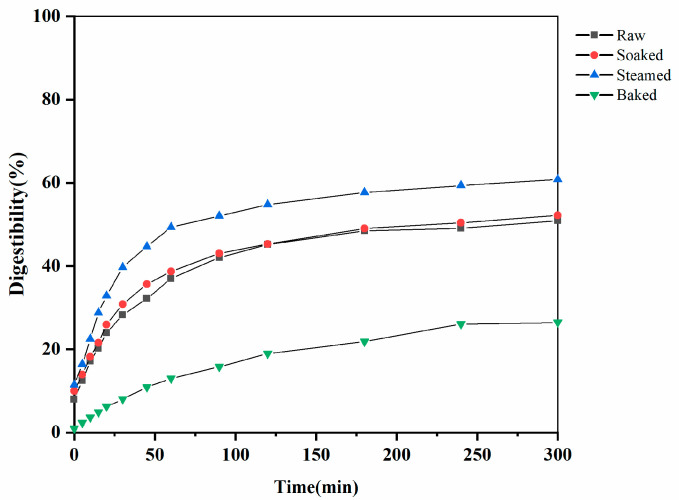
Starch digestion curves of M1 with different processing treatments.

**Figure 7 foods-11-00212-f007:**
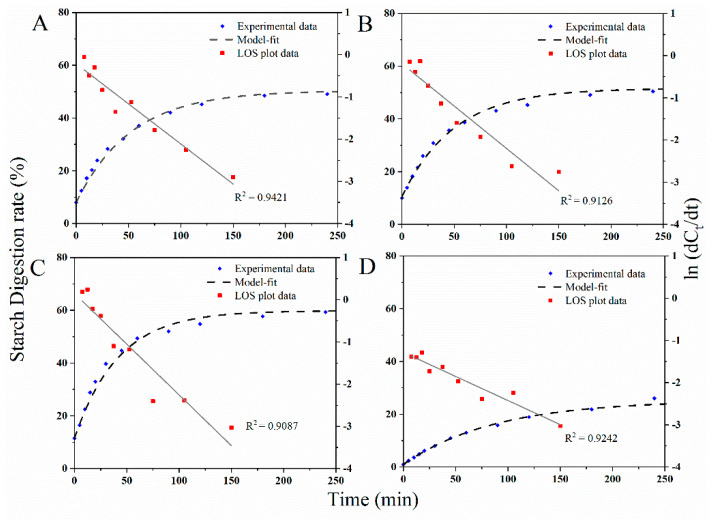
Starch digestion curves, LOS plots, and model-fit curves of M1 with different processing treatments. (**A**–**D**) represents raw millet, soaked millet, steamed millet, and baked millet, respectively.

**Table 1 foods-11-00212-t001:** The parameters of molecular size distribution, chain length distribution, and LOS linear fitting parameters of raw and different processed millets.

Millet Cultivar/Processing Methods		$\bar{R_{h}}$	Am	AP CLD					AM CLD		LOS	
				6 < *X* ≤ 12	13 < *X* ≤ 24	24 < *X* ≤ 36	36 < *X* ≤ 100	100 < *X* ≤ 1000	1000 < *X* ≤ 5000	5000 < *X* ≤ 20,000	*k* min^−1^	*C*_∞_%
M1	Raw	11.9 ± 0.8 ^d^	34.3 ± 2.2 ^a^	16.2 ± 1.7 ^gh^	26.9 ± 0.7 ^g^	9.6 ± 0.3 ^ef^	13.7 ± 0.1 ^def^	21.3 ± 0.3 ^ab^	11.1 ± 0.2 ^ab^	0.9 ± 0.0 ^a^	0.019 ± 0.001 ^cd^	50 ± 0.2 ^cd^
	Soaked	11.4 ± 0.9 ^d^	34 ± 1.3 ^b^	16.7 ± 0.5 ^hi^	26.5 ± 0.2 ^g^	9.6 ± 0.1 ^ef^	14.2 ± 0.1 ^fg^	20.9 ± 0.1 ^a^	10.6 ± 0.0 ^a^	1.4 ± 0.1 ^a^	0.0201 ± 0.005 ^d^	51 ± 0.4 ^d^
	Steamed	9.7 ± 0.7 ^c^	36.6 ± 1.5 ^d^	17 ± 0.2 ^i^	24.8 ± 1.0 ^e^	9 ± 0.4 ^e^	12.6 ± 0.2 ^c^	23.6 ± 0.0 ^c^	11.4 ± 0.0 ^b^	1.5 ± 0.1 ^ab^	0.024 ± 0.004 ^a^	60 ± 1.0 ^fg^
	Baked	7.7 ± 0.1 ^a^	63 ± 5.6 ^h^	8.9 ± 0.3 ^b^	14.3 ± 0.7 ^b^	5.1 ± 0.6 ^b^	8.7 ± 0.5 ^a^	42 ± 1.3 ^f^	18.4 ± 0.2 ^f^	2.5 ± 0.0 ^c^	0.0114 ± 0.002 ^a^	25 ± 0.5 ^a^
M2	Raw	13.3 ± 0.1 ^e^	33.5 ± 1.6 ^b^	14.8 ± 0.3 ^e^	25.5 ± 1.3 ^f^	9.0 ± 0.7 ^e^	13.8 ± 2.1 ^efg^	23.2 ± 1.2 ^c^	12.5 ± 0.1 ^c^	1.3 ± 0.0 ^a^	0.016 ± 0.002 ^bc^	58 ± 1.2 ^efg^
	Soaked	13.2 ± 0.8 ^e^	34 ± 0.9 ^b^	14.6 ± 0.1 ^e^	25.6 ± 0.9 ^f^	9.0 ± 0.3 ^e^	13.2 ± 1.2 ^d^	23.0 ± 0.9 ^c^	13.3 ± 0.6 ^d^	1.2 ± 0.1 ^a^	0.0156 ± 0.0005 ^bc^	62 ± 1.9 ^g^
	Steamed	10.2 ± 0.5 ^c^	40.9 ± 2.7 ^f^	13.6 ± 0.1 ^d^	23.5 ± 2.3 ^cd^	8.2 ± 0.1 ^d^	13.8 ± 0.9 ^efg^	25.9 ± 0.7 ^d^	13.8 ± 0.4 ^d^	1.2 ± 0.2 ^a^	0.0208 ± 0.003 ^d^	82 ± 2.5 ^h^
	Baked	8.9 ± 0.4 ^b^	58.3 ± 2.2 ^g^	9.6 ± 0.2 ^c^	15.6 ± 2.0 ^d^	6.0 ± 0.1 ^c^	10.5 ± 1.2 ^b^	39.8 ± 1.8 ^e^	16.3 ± 0.2 ^e^	2.2 ± 0.0 ^bc^	0.013 ± 0.004 ^ab^	42 ± 1.2 ^b^
M3	Raw	14.9 ± 1.1 ^f^	32.5 ± 0.7 ^b^	15.7 ± 1.1 ^fg^	27.2 ± 2.9 ^c^	9.9 ± 0.4 ^f^	13.7 ± 1.2 ^def^	21.0 ± 0.6 ^a^	11.6 ± 0.1 ^b^	0.9 ± 0.1 ^a^	0.0187 ± 0.0015 ^cd^	55 ± 0.7 ^def^
	Soaked	14.8 ± 0.7 ^f^	32.3 ± 0.3 ^b^	15.8 ± 0.8 ^fg^	26.8 ± 1.8 ^a^	9.9 ± 0.2 ^f^	14.3 ± 1.0 ^g^	20.9 ± 2.2 ^a^	11.6 ± 0.1 ^b^	0.9 ± 0.0 ^a^	0.0194 ± 0.001 ^cd^	54 ± 0.6 ^de^
	Steamed	11.7 ± 0.2 ^d^	35.8 ± 1.2 ^c^	15.2 ± 0.6 ^ef^	26.2 ± 1.9 ^c^	9.4 ± 1.0 ^ef^	13.5 ± 2.1 ^de^	21.9 ± 1.2 ^b^	12.5 ± 0.0 ^c^	1.3 ± 0.3 ^a^	0.0276 ± 0.002 ^e^	80 ± 1.2 ^h^
	Baked	9.9 ± 0.2 ^c^	67.9 ± 2.4 ^i^	7.1 ± 0.2 ^a^	11.9 ± 0.7 ^a^	4.3 ± 0.5 ^a^	8.7 ± 1.9 ^a^	45.4 ± 1.4 ^g^	19.1 ± 0.2 ^g^	3.4 ± 0.0 ^d^	0.0103 ± 0.0007 ^a^	45 ± 0.5 ^bc^

AP: Amylopectin, AM: Amylose, $\bar{R_{h}}$: the average R_h_, Am: Amylose content, *k*: starch digestion rate coefficient, *C*_∞_: the estimated percentage of starch digested at the end-point of the reaction. Values of the same parameter with a different superscript letter in a column indicate significant differences (*p* < 0.05).

## Data Availability

Not applicable.

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
