# Peer review of "Effects of Processing on Starch Structure, Textural, and Digestive Property of “Horisenbada”, a Traditional Mongolian Food"

_foods, 2022, doi:10.3390/foods11020212_

Round 1

Reviewer 1 Report

The authors have studied the impact of various cooking procedures of different cultivars of millet on the morphology, structure and digestion of the resulting products.

The English language is not good enough for a publication in Foods. The manuscript should be read and corrected by an English-speaking person. This would prevent several ambiguities. Strangely, the introduction is quite correct.

- Line 30: why is this reference not properly numbered?

- Line 42: what are the "diverse solutions"? Please, be precise.

- Line 58: what is the meaning of "larger and less dispersive starch granules"? Please rephrase.

- Line 62: to verify

- Line 95-98: First, the English is poor. Please rephrase this section. Second, the conditioning of the specimens for XRD is not described. It is well-known that the XRD profile and calculated relative crystallinity of starchy products significantly depends on the sample hydration. The authors should indicate if the specimen was dry or if it had been conditioned at a given relative humidity. Third, gelatinized starch is not stable over time and a fraction of the molecules can partially recrystallize, which would affect the relative crystallinity.

- Line 102: kV

- Line 115 and in the section: mL and µL (please separate the unit from the numbers).

- Line 151: why "Apparently"? Please delete.

- Line 153-154: please do not use "it is obvious". Not scientific and unnecessary.

- Line 171: please rephrase "analogical".

- Figure 3: please indicate the unit for the diffraction angles.

- Line 204: spherical shape

- Line 205: starch granular morphology

- Line 207-208: why was gelatinization simply not considered?

- Figure 4: Since the images are rather small, it is very difficult to see what the authors would like the reader to see. Close-ups might be useful, in particular for the regions that are outlined. In addition, the dark technical bar below each image is unreadable so they should be removed and proper scale bars should be indicated on all images.

- Section 3.4.: Again, I guess mechanical properties should depend on the hydration of the specimens. Nothing is indicated in section 2.7.

- Line 231: Delete "apparently".

I am somewhat frustrated by this manuscript. The experimental results are interesting but, while comparisons between specimens are provided, the article does not bring any discussion or explanation on the origin of the differences. It is merely a descriptive study but there is very little science related to starch. I disagree with the statement in line 280: there are no links clearly identified on the impact of the different treatments, especially in relation with the water content of the specimens. The fact that gelatinization occurs is not particularly surprising. That the Rh varies is more interesting and that the chain length distribution varies so much is indeed rather puzzling. Why would "new glycosidic bonds [be formed] under high temperature, resulting in generation of some non-linear structures"? Would only heating promote the formation of glycosidic bonds? Is it documented? Has it been demonstrated by baking pure linear amylose as a control? At this temperature, I would rather expect a depolymerization of the chains, possibly by an oxidation effect. I cannot say for sure but the authors should at least further discuss this effect in more details.

The style of *all* references should be homogenized considering the requirements of Foods. It is not the case: some journals are abbreviated (sometimes improperly like in Ref. #3 – see ref. 24), others not; titles and journal names should not be capitalized. Please take some time to read and check the references *one by one*.

Please identify properly the Supplementary data file: article title and authors.

My recommendation is that significant revisions must be done.

Reviewer 2 Report

General overview.

Manuscript foods-1515991 titled “Effects of processing on starch structure, textural, and digestive

property of “Horisenbada”, a traditional mongolia food”, is an evaluation of the effects of processing on the millet “Horisenbada” starch. In general, the work is very interesting however, the real limits of this studies are mainly due to the statistical elaboration and the result discussion is poor

Major comments.

  • Is the statistic elaboration appropriate??? Parametric ANOVA is not the best statistical test for small dataset. I suggest to apply non parametric variances analyses.
  • more discussions in particular of the variation of the starch characteristic would be appreciated

Minor comments.

Line 20-21 Is not clear the relation of this work and health beneficial (add in discussion section);

Line 29 is not clear if whole millet is natural rich in essential nutrient or was added the nutrient “enriched” to the millet;

Line 61-63 the aim is very poor add more information about the scope of the work otherwise it would seem a poorly innovative work;

Line 95 add information about diffractometer (brand, model, etc etc);

Line 125 add information about ELISA instrument used (brand, model, etc etc);

Check that the number of figures is correct both in the text and in the captions

Reviewer 3 Report

The manuscript presents interesting experiment, however many details are not clear and have to clarified (details below). The style of the manuscript (English) requires thorough review.

Line 25-26 I think addition of more detail regarding Horisenbada and it’s processing would be beneficial for most readers

Line 29-30 not clear how the second part of the sentence is linked with the first one

Line 30-31 the effect were not stated earlier

Line 41-42 style

Line 56-59 the statement is not quite clear, and obvious as indicated

Line 76-78, the process should be described with more details, especially regarding amount of water during steam cooking (was the water discarded after soaking?) if not was it discarded prior baking? Moisture content of millets prior each treatment should be also evaluated.

Line 84 PSS is a brand not a model of chromatograph

Line 87 sentence not clear

Line 96 just degrees not Celsius

Line 102 kV

Line 112 not for, but through

Line 125 ELIASA?

Line 150 why molecular weight is denoted as Rh? Usually Rh is denoted as hydrodynamic radius and is a SEC parameter obtained in systems with triple detection. The unit is also related to Rh. It is not clear and should be clarified in the whole SEC discussion

Line 154 in my opinion the amount of amylopectin does not change it the solubility of the sample and possible changes in Rg/Rh the alter the results

Line 163 table 1 lacking explanatory notes

Line 168 elsewhere not cited

Line 177-178 this is possible but at relatively high moisture content (it was not shonw)

Line 209- the microphotogrpahs should be in better resolution, moreover raw samples A anc C look like not correct (partially gelatinized)

Line 220 sole soaking cannot cause gelatinization, please rephrase

Line 224 relatively small differences between treatments of samples M3 should be discussed

Line 254 I think it would be better if the scaling was the same on all graphs for comparison

Line 262 – based on the data the crystalline structure only decreased partially (around 50%)

Round 2

Reviewer 1 Report

The authors considered the comments and questions from the referees but only partially corrected the manuscript accordingly. Several corrections of the English must be done. I do not have the impression that the revised version was corrected by an English-speaking person. Even using automatic grammar correctors should be sufficient to write proper English and prevent ambiguities in the meaning.

I have just indicated a few errors below but there are more that must be corrected before publication (not really the job of a referee).

Line 58: the surface of millet grains

Line 59: Beside the sensory

Line 114: was used

Line 116: How was the moisture content equilibrated to 10%?

Section 2.7: Please separate the units form the numbers with a space

Section 2.8: In the whole section, use mL and µL like in section 2.4 (please separate the unit from the numbers).

Line 133: Millet grains were

Line 196: please define X

Line 206: please rephrase

Line 215: radius

Line 240: was polygonal

Line 242: be associated

Line 243: rephrase the whole sentence.

Line 252: Delete "Apparently"

Line 307: rephrase the whole sentence.

Figure 1: The x-axis is Rh so the legend should not say that these are the molecular weight distributions.

Figure 2: Please indicate a clear information on the nature of the x-axis data: meaning of X and unit.

Please identify properly the Supplementary Material file by adding the article title and authors at the beginning of page 1.

Author Response

The authors considered the comments and questions from the referees but only partially corrected the manuscript accordingly. Several corrections of the English must be done. I do not have the impression that the revised version was corrected by an English-speaking person. Even using automatic grammar correctors should be sufficient to write proper English and prevent ambiguities in the meaning.

Response: We have now modified our language by using a commercial language editing service. The language modifications were also labeled in the main text.

I have just indicated a few errors below but there are more that must be corrected before publication (not really the job of a referee).

  1. Line 58: the surface of millet grains

Response: Thanks for your careful review. The relevant descriptions have been revised.

Line 57-58: “It has been proved baking enhanced the flavor by browning the surface of millet grains …”.

  1. Line 59: Beside the sensory

Response: Thanks for your careful review. The text has been modified.

Line 59-60: “In addition to changing the flavor properties, …”.

  1. Line 114: was used

Response: Thanks for your suggestion. The words have been replaced.

Line 108-109: “The X-ray diffractometer (D2 PHASER, Bruker AXS GMBH, Karlsruhe, Germany) with Cu Kα radiation (λ = 0.154) was used to record XRD patterns.”.

  1. Line 116: How was the moisture content equilibrated to 10%?

Response: Thanks for your question. Details on the samples determined by XRD have been added.

Line 110-112: “the moisture content of the samples was equilibrated to about 10% by standing at room temperature overnight.”.

  1. Section 2.7: Please separate the units form the numbers with a space

Response: Thanks for your careful review. The text has been revised.

  1. Section 2.8: In the whole section, use mL and L like in section 2.4 (please separate the unit from the numbers).

Respone: Thanks for your review. The whole section has been revised.

  1. Line 133: Millet grains were

Response: Thanks for your careful review. The text has been revised.

Line 128: “Millet grains were ground and sifted through 80 mesh.”.

  1. Line 196: please define X

Response: Thanks for your comment. The information about X has been defined.

Line 190-191: “Starch CLDs, w(logX), obtained from the DRI signal was plotted against DP X.”.

  1. Line 206: please rephrase

Response: Thanks for your suggestion. The sentence has been rephrased.

Line 203-204: “Compared with raw and other processed millet, the chain length distribution of baked millet starch had a higher proportion of chains with X-100~1000 and X-1000~5000.”.

  1. Line 215: radius

Response: Thanks for your careful review. The spelling error has been corrected.

  1. Line 240: was polygonal

Response: Thanks for your review. This sentence has been revised.

Line 236-237: “whereas the granular structure of raw, soaked, and steamed millet starch of M2 and M3 was polygonal shapes.”.

  1. Line 242: be associated

Response: Thanks for your careful view. The words have been revised.

  1. Line 243: rephrase the whole sentence.

Response: Thanks for your suggestion. The whole sentence has been revised.

Line 239-241: “For three millet cultivars, the baked millet starch granules were partially destroyed and aggregated into irregular lumps with a closely packed granule structure.”.

  1. Line 252: Delete "Apparently"

Response: Thanks for your suggestion. The word has been deleted.

  1. Line 307: rephrase the whole sentence.

Response: Thanks for your comment. The whole sentence has been revised.

Line 304-306: “For details, the steamed millet showed the rapid digestive property, which was consistent with the highest k and C values. The increased millet digestion may be caused by the gelatinization of millet starch by steaming process.”.

  1. Figure 1: The x-axis is Rh so the legend should not say that these are the molecular weight distributions.

Response: Thanks for your question, Rh, the hydrodynamic radius, is the parameter for characterizing molecular size of starch. It has widely used to parameterize molecular weight distributions.

  1. Figure 2: Please indicate a clear information on the nature of the x-axis data: meaning of X and unit.

Response: Thanks for your suggestion. The definition of X has been added.

Line 216: “X represented the DP value of an individual chain.”.

  1. Please identify properly the Supplementary Material file by adding the article title and authors at the beginning of page 1.

Response: It has been revised accordingly.

Reviewer 2 Report

no comment

Author Response

Thanks for your recommendation.

Reviewer 3 Report

The manuscript has been improved substantially and can be recommended for publication.

Author Response

Thanks for your recommendation.